# Optical spin-state polarization in a binuclear europium complex towards molecule-based coherent light-spin interfaces

Kuppusamy Senthil Kumar [1,2,8✉], Diana Serrano[3,8✉], Aline M. Nonat[4], Benoît Heinrich[1], Lydia Karmazin [5], Loïc J. Charbonnière [4], Philippe Goldner [3✉] & Mario Ruben [2,6,7✉]

The success of the emerging field of solid-state optical quantum information processing (QIP) critically depends on the access to resonant optical materials. Rare-earth ion (REI)-based molecular systems, whose quantum properties could be tuned taking advantage of molecular engineering strategies, are one of the systems actively pursued for the implementation of QIP schemes. Herein, we demonstrate the efficient polarization of ground-state nuclear spins—a fundamental requirement for all-optical spin initialization and addressing—in a binuclear Eu(III) complex, featuring inhomogeneously broadened $^5D_0 \rightarrow {}^7F_0$ optical transition. At 1.4 K, long-lived spectral holes have been burnt in the transition: homogeneous linewidth ($\Gamma_h$) = 22 ± 1 MHz, which translates as optical coherence lifetime ($T_{2opt}$) = 14.5 ± 0.7 ns, and ground-state spin population lifetime ($T_{1spin}$) = 1.6 ± 0.4 s have been obtained. The results presented in this study could be a progressive step towards the realization of molecule-based coherent light-spin QIP interfaces.

[1] Institut de Physique et Chimie des Matériaux de Strasbourg (IPCMS), CNRS-Université de Strasbourg, Strasbourg, France. [2] Institute of Nanotechnology, Karlsruhe Institute of Technology (KIT), Eggenstein-Leopoldshafen, Germany. [3] Institut de Recherche de Chimie Paris (IRCP), Université PSL, Chimie ParisTech, CNRS, Paris, France. [4] Equipe de Synthèse pour l'Analyse, IPHC, UMR 7178, CNRS-Université de Strasbourg, ECPM, Strasbourg, France. [5] Service de Radiocristallographie, Fédération de Chimie Le Bel FR2010 CNRS-Université de Strasbourg, Strasbourg, France. [6] Institute for Quantum Materials and Technologies (IQMT), Karlsruhe Institute of Technology (KIT), Eggenstein-Leopoldshafen, Germany. [7] Centre Européen de Sciences Quantiques, Institute de Science et d'Ingénierie Supramoléculaire (ISIS), Université de Strasbourg, Strasbourg, France. [8] These authors contributed equally: Kuppusamy Senthil Kumar, Diana Serrano. ✉email: senthil.kuppusamy2@kit.edu; diana.serrano@chimieparistech.psl.eu; philippe.goldner@chimieparistech.psl.eu; mario.ruben@kit.edu

Quantum information processing (QIP) schemes, such as quantum computing, quantum storage, and quantum communication, use the quantum nature of materials to process and manipulate information[1–4]. In QIP, a dramatic improvement in computation time and secure data transmission can be achieved by creating superposition states with long coherence lifetimes ($T_2$)[5,6]. Environmental fluctuations due to lattice phonons, molecular vibrations, and magnetic moments reduce the coherence lifetime of a superposition state[7,8]. Thus, to achieve superposition states with coherence lifetimes suitable for realistic applications, the coherent states must be placed in a non-fluctuating environment.

Optical qubit operations can be performed with systems featuring narrow linewidth optical transitions. Color centers in diamond[4], rare-earth ions (REIs) doped in host matrices[2,9–13], and luminescent organic dye molecules (dye impurity), featuring extremely narrow and stable luminescent lines[14–17], are suitable systems to implement optical qubit operations. REI-doped systems are particularly well-suited for optical QIP applications due to the following intrinsic properties. REIs feature long optical coherence lifetimes ($T_{2opt}$), because the $4f$–$4f$ optical transitions, covering the whole visible and infrared spectral range, are well-shielded from the surrounding environment by the outer $5s$ and $6p$ orbitals. The nuclear spin (I) containing REI-isotopes enable the creation of nuclear spin superposition states with long spin coherence lifetimes ($T_{2spin}$), useful for storing quantum states[11,18,19]. Importantly, the exceptionally good optical coherence lifetimes associated with the $4f$–$4f$ transitions allow for coherent optical addressing and manipulation of nuclear spin states[20–22].

Non-Kramers REIs with an even number of f electrons—for example, Eu(III), Pr(III), or Tm(III)—embedded in a matrix with low average magnetic moments have been extensively studied for the implementation of QIP schemes[23]. The $^5D_0 \rightarrow {}^7F_0$ transition of Eu(III) is of particular interest because the $^5D_0 \rightarrow {}^7F_0$ transition is an induced electric dipole transition[24] and is largely unaffected by the magnetic field fluctuations arising from the surrounding environment, thereby long optical coherence lifetimes are associated with the transition.

The QIP utility of Eu(III)-doped ceramics, powders, crystals, and nanoparticles—featuring $^5D_0 \rightarrow {}^7F_0$ transition—has been elucidated[2,25–28]. However, it is difficult to tailor-make such systems with desirable optical properties by means of chemical and physical manipulations, limiting the utility of such materials for QIP applications. On the other hand, the optical properties of molecular Eu(III) complexes can be easily tuned by ligand field and molecular energy level engineering approaches, as previously exploited for applications, such as temperature sensing and bioimaging[29,30]. Further, by synthesizing isotopically enriched nuclear spin-free ligands, minimization of phonon and nuclear spin-mediated relaxation pathways could be obtained, thereby high quantum efficiencies and long $T_{2opt}$ could be achieved. Importantly, Eu(III) complexes with isotopically pure emitting centers, for example, $^{151}Eu$ or $^{153}Eu$, both with I = 5/2, can be synthesized via isotopologues coordination chemistry[31,32], enabling the use of nuclear spin states as storage states for QIP applications.

Frequency domain techniques such as transient spectral hole burning (SHB) have been used to optically probe superposition state lifetimes in molecular[33,34] and REI-doped systems; for example, Eu(III):Y$_2$O$_3$[26]. Transient spectral holes appear as dips in absorption or excitation spectrum when a portion of active ions do not show ground-state absorption anymore after optical pumping and for a certain time duration. The utility of transient SHB for measuring coherence lifetimes relies on the following factors, (i) existence of inhomogeneously broadened transitions

with $\Gamma_{inh} \gg \Gamma_h$, where $\Gamma_{inh}$ and $\Gamma_h$ are the optical inhomogeneous and homogeneous linewidths, respectively; (ii) $\Gamma_h \gg \Gamma_{laser}$, where $\Gamma_{laser}$ is the excitation laser linewidth; and (iii) $\Gamma_h < \tau^{-1}$ and/or $T_{1spin}^{-1}$, for SHB in the excited-state and/or in the ground-state nuclear spin levels, where $\tau$ and $T_{1spin}$ are the optical excited-state lifetime and the ground-state spin population lifetime, respectively. In these conditions, the hole width ($\Gamma_{hole}$) is equal to $2\Gamma_h$, with $\Gamma_h$ inversely proportional to the optical coherence lifetime ($\Gamma_h = 1/\pi T_{2opt}$).

Reports on SHB in REIs are rather limited to REIs dispersed in matrices, such as Er$^{3+}$:$^7$LiYF$_4$, Nd$^{3+}$:YVO$_4$, Eu$^{3+}$:Y$_2$SiO$_5$, and Eu$^{3+}$:Y$_2$O$_3$[23]. To the best of our knowledge, transient SHB in a molecular REI system—especially in an Eu(III) complex—is yet to be reported. Therefore, performing SHB studies in REI-based molecular systems to elucidate a light-mediated contol over nuclear spin states is a starting point towards the realization of REI molecules-based photonic quantum materials[3,35].

In this study, we demonstrate transient SHB (see "Methods," Supplementary Section 2, and Supplementary Figures 1–4 for experimental details) in the inhomogeneously broadened $^5D_0 \rightarrow {}^7F_0$ optical transition of a binuclear molecular Eu(III) complex—[Eu$_2$Cl$_6$(4-picNO)$_4$($\mu_2$-4-picNO)$_2$]·2H$_2$O ([Eu$_2$]); 4-picNO stands for 4-picoline-N-oxide. The measured hole width yields a homogeneous linewidth of $22 \pm 1$ MHz, which corresponds to $T_{2opt}$ of $14.5 \pm 0.7$ ns. A hole decay time of $1.6 \pm 0.4$ s is observed, consistent with nuclear spin relaxation, confirming the utility of the SHB method used here to optically polarize the nuclear spin states in the [Eu$_2$] complex.

## Results

**Design considerations**. To implement optical QIP schemes, a molecular Eu(III) complex should feature a long coherence time, photostability, stability to be handled at ambient conditions, and a reasonable luminescence quantum yield. Importantly, a low symmetry environment around the Eu(III) center should be satisfied to observe the $^5D_0 \rightarrow {}^7F_0$ transition.

R-pyridine-N-oxide (RPYNO; R = H, CH$_3$, and CF$_3$) ligands are suitable for the development of Eu(III)-complexes for QIP applications because such ligands feature nuclear spin-free oxygen donor sites and sensitize the Eu(III)-luminescence via antenna effect[24,36,37]. Moreover, the $\mu_2$-coordination mode of RPYNO ligands facilitates the formation of low-coordinate binuclear Eu complexes with desired ligand field symmetry around the Eu(III) center, promoting the appearance of $^5D_0 \rightarrow {}^7F_0$ transition. On the downside, RPYNO ligands feature nuclear spin containing hydrogen ($^1$H; abundance >99.98% and I = 1/2) atoms in their backbone, which reduce the coherence lifetime. Therefore, the number of multiatomic ligands—containing hydrogen atoms—coordinating with Eu(III) should be reduced to improve the coherence lifetime.

A reduced number of hydrogen-containing ligands around Eu (III) could be achieved employing monoatomic halide-based ligands—such as Cl—as coordinating ligands in view of their heavier nature compared to hydrogen. On the other hand, Cl$^-$ ligands do not sensitize the Eu(III) luminescence and are composed of nuclear spin carrying isotopes—$^{35}$Cl and $^{37}$Cl, both with I = 3/2. Remarkably, the gyromagnetic ratio ($\gamma$) of nuclear spin isotopes of chlorine—$\gamma = 2.62 \times 10^7$ and $2.18 \times 10^7$ rad s$^{-1}$ T$^{-1}$ for $^{35}$Cl and $^{37}$Cl, respectively—is about ten times smaller than the gyromagnetic ratio of hydrogen ($26.75 \times 10^7$ rad s$^{-1}$ T$^{-1}$). Thus, the use of Cl$^-$ ligands is advantageous to reduce decoherence pathways arising from magnetic interactions.

**Preparation and X-ray structure analysis of [Eu$_2$]**. Based on the above rationale, we have synthesized the binuclear [Eu$_2$] by taking

advantage of the luminescence sensitization efficiency of a RPYNO ligand—4-picoline N-oxide (4-picNO)—and heavy and monoatomic nature of $Cl^-$ ligand. Overall, a trade-off between Eu (III) luminescence sensitization and decoherence reduction have been attempted.

The [$Eu_2$] complex was prepared by treating the commercially available $EuCl_3 \cdot 6H_2O$ and 4-picNO in 1:3 ratio in water followed by recrystallization of the crude reaction mixture from ethanol-ethyl acetate solvent mixture (see Supplementary Section 2.3 and Supplementary Figure 5 for more details). The synthesis of the [$Eu_2$] complex is reproducible, and the method employed to prepare the complex was successfully extended to prepare a series of binuclear Ln(III) complexes—[$Ln_2Cl_6$(4-picNO)$_4$($\mu_2$-4-pic-NO)$_2$]·$2H_2O$ (Ln = Tb, Gd, Dy, Er, and Ho)—establishing the versatility of the procedure adopted. Since [$Gd_2Cl_6$(4-picNO)$_4$($\mu_2$-4-picNO)$_2$]·$2H_2O$ ([$Gd_2$]) was used to estimate the triplet energy ($^3\pi\pi$) of the 4-picNO ligand, the preparation and structural characterization of [$Gd_2$] is presented in the Supplementary Sections 2.3 and 3. X-ray crystal structure and the associated photophysical and magnetic properties of the remaining complexes—(Ln = Tb, Dy, Er, and Ho)—will be reported elsewhere.

X-ray crystallographic analysis revealed the binuclear structure of [$Eu_2$], as shown in Fig. 1a, b; see Supplementary Fig. 6 and Supplementary Table 1 for packing in the crystal lattice and crystallographic data, respectively. The complex crystallized in the centrosymmetric $P\bar{1}$ space group, belonging to the triclinic crystal system. The neutral Eu(III) dimer is composed of six 4-picNO ligands and six chloride ligands. The $O_4Cl_3$ coordination environment around each Eu(III) ion is best described as pentagonal bipyramidal with a continuous shape measure (CShM)[38] of 1.513, as shown in Fig. 1b. In [$Eu_2$], the equatorial positions (edges) of each pentagonal bipyramid are occupied by two monodentate 4-picNO ligands in trans fashion, one chloride ligand, and two $\mu_2$-4-picNO ligands. The axial positions of each pentagonal bipyramid are occupied by the remaining chloride ligands. The intramolecular Eu···Eu distance is 4.273(4) Å. A good match between the powder X-ray diffraction (PXRD) and single-crystal XRD (SCXRD) patterns (Supplementary Figure 7) and the similar unit cell parameters (Supplementary Table 2) obtained from the indexing of the PXRD and SCXRD data unambiguously prove the phase purity of the crystalline material utilized for photophysical and SHB studies.

**Photophysical studies**. The binuclear complex, [$Eu_2$], exhibited sensitized (Fig. 2a) Eu(III)-based emission upon excitation of 4-

picNO-based transition centered ~330 nm. Among the $^5D_0 \rightarrow {}^7F_J$ ($J = 0$–6) transitions observed (Fig. 2b), the $^5D_0 \rightarrow {}^7F_0$ transition is of interest to this study. The sharp and nondegenerate (vide infra) nature of the $^5D_0 \rightarrow {}^7F_0$ transition is in agreement with identical environments around the two Eu(III) centers in the dimer. However, since the two sites have different orientations within the molecule, they are inequivalent and they can be differentiated; for example, upon application of an external magnetic field.

At room temperature, luminescence lifetime ($\tau_{obs}$) of 822 μs (Supplementary Figure 11b) and total emission quantum yield ($Q_{tot}$) of 38 ± 6% were determined for [$Eu_2$]. See Supplementary Section 3.3.1 for a detailed analysis of the sensitized emission process in [$Eu_2$].

**Low-temperature high-resolution spectroscopy and SHB**. The $^5D_0 \rightarrow {}^7F_0$ transition of Eu(III) is of particular interest for SHB studies. The photoluminescence excitation (PLE) spectrum, recorded at 1.4 K, of the $^5D_0 \rightarrow {}^7F_0$ optical transition in [$Eu_2$] (Fig. 2c) is composed of several partially resolved peaks. The most prominent appeared at 580.185 nm (vac.) with a full-width at half-maximum (FWHM) of 0.06 nm (50 GHz or 1.7 cm$^{-1}$). This is comparable to inhomogeneous linewidths observed in Cr(III) compounds[39]. However, the linewidth is larger than optical inhomogeneous linewidths observed in most inorganic crystals, in the few GHz range[40]. A $\tau_{obs} = 880$ μs was obtained from a single exponential fit of the $^5D_0$ fluorescence decay measured at 20 K under resonant excitation at 580.185 nm (Fig. 2d).

The complexity of this absorption spectrum for a transition taking place between two singlet electronic levels, and Eu(III) ions placed in an identical ligand field environment in each [$Eu_2$], raises the question of the origin of the different peaks. Selective excitation of the different peaks (Fig. 2c) reveals differences in the low-temperature PL emission spectra and observed lifetimes (Supplementary Figure 13). This indicates the existence of subsite structures[41,42] along with main sites (580.185 nm) in the crystal lattice of [$Eu_2$], tentatively attributed to a defective crystal lattice. The single exponential character of the decay profiles, obtained after exciting the different peaks in Fig. 2c, reveals the absence of interactions between Eu(III) centers in different subsites.

Homogeneous linewidths are of major importance to evaluate the potential of a system for optical QIP applications. In particular, narrow homogeneous linewidths are required for optical quantum storage and subsequent quantum coherence transfer to the nuclear spin states. Homogeneous linewidth values are, however, difficult to access from the inhomogeneously broadened lines of REI-based systems by classical absorption and fluorescence techniques. To circumvent this issue, we have used SHB[43] and probed the optical homogeneous linewidth of $^5D_0 \rightarrow {}^7F_0$ transition associated with [$Eu_2$] and demonstrated selective optical addressing of the ground-state nuclear spin levels in the complex (see "Methods," Supplementary Section 2, and Supplementary Figures 2–4 for details). The applied optical pumping continuously transfers a resonant ensemble of Eu(III) molecules to the excited state (Fig. 3b). Once in the excited-state, three decay channels are possible, because Eu(III) presents three doubly degenerate ground-state nuclear spin levels (±5/2, ±3/2, and ±1/2; for $^{151/153}$Eu, I = 5/2). Ions decaying back to the initial-state will be immediately pumped back to excited states; the ions relaxing to other nuclear spin states are off-resonant with the excitation laser; therefore, the population is stored in these states until spin relaxation occurs.

As the pumping proceeds, the initial spin states are progressively emptied, and the population is transferred to different spin levels (Fig. 3b). Scanning the laser over the optical

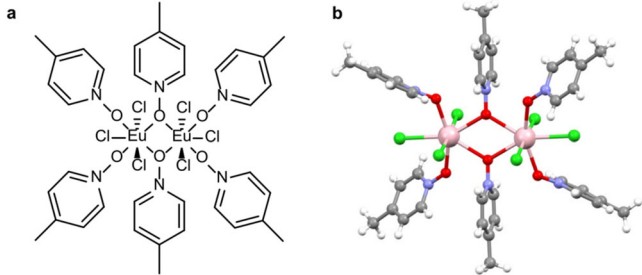

**Fig. 1 Structure of [$Eu_2Cl_6$(4-picNO)$_4$($\mu_2$-4-picNO)$_2$]·$2H_2O$. a** Molecular structure of the complex showing the ligands coordinating with the Eu(III) centers. The complex was prepared by treating $EuCl_3 \cdot 6H_2O$ with 4-picNO ligand dissolved in water followed by a recrystallization step from ethanol (EtOH)/ethyl acetate (EtOAc) solvent mixture. **b** X-ray crystal structure of the complex. Coordination geometry around each Eu(III) center of the complex is best described as pentagonal bipyramidal. The co-crystallized water molecules are omitted for clarity. Color code: C, gray; Cl, green; Eu, pink; H, white; N, blue; O, red.

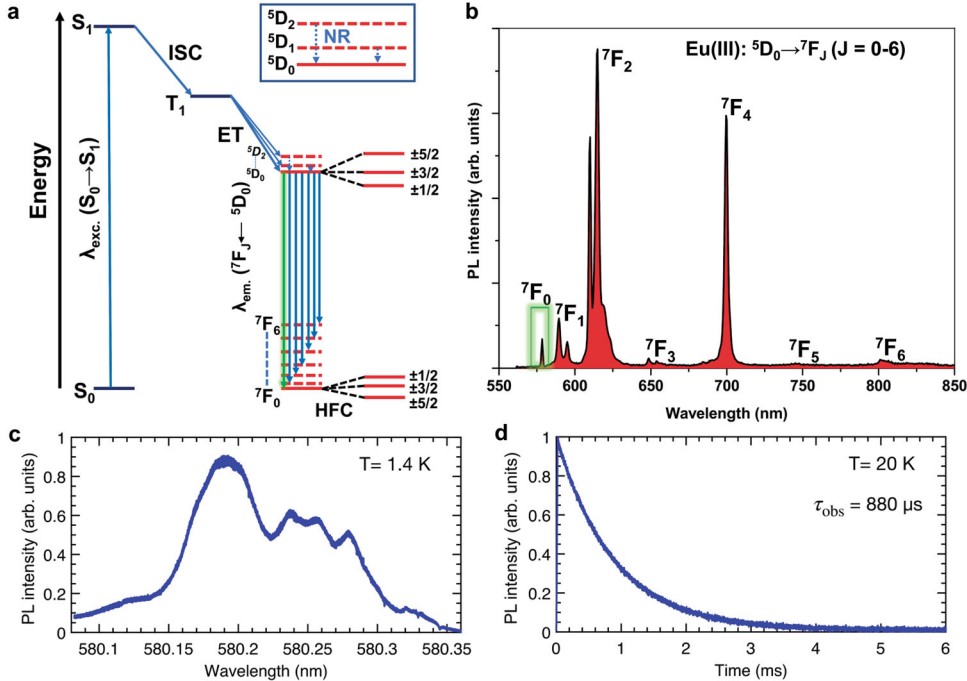

**Fig. 2 Photophysical properties of [Eu$_2$] in the solid-state. a** Mechanism of PL sensitization in Eu(III) complexes: the Eu(III)-based $^5D_J$ ($J = 0$–2) receiving levels are populated after a series of excitation, intersystem crossing (ISC), and $T_1 \rightarrow {}^5D_J$ energy transfer (ET) processes. The co-populated excited $^5D_2$ and $^5D_1$ levels non-radiatively (NR) relax to $^5D_0$ level (inset); radiative relaxation of $^5D_0$ level to ground $^7F_J$ ($J = 0$–6) crystal field levels manifests as line-like luminescence. The $^5D_0 \rightarrow {}^7F_0$ transition is suited for QIP applications due to its narrow linewidth and long coherence lifetimes of the nuclear spin states—±5/2, ±3/2, ±1/2 ($I = 5/2$ for $^{151}$Eu/$^{153}$Eu)—associated with the $^7F_0$ ground-state level. **b** Photoluminescence spectrum showing the $^5D_0 \rightarrow {}^7F_J$ ($J = 0$–6) transitions in the visible and near-IR range ($\lambda_{exc.} = 330$ nm). **c** Photoluminescence excitation (PLE) spectrum of the $^5D_0 \rightarrow {}^7F_0$ transition of [Eu$_2$] measured at 1.4 K. An inhomogeneous linewidth ($\Gamma_{inh}$) of 50 GHz is calculated for the main peak centered at 580.185 nm (vac.) **d** Luminescence decay of the $^5D_0$ excited-state of [Eu$_2$] measured at 20 K under resonant excitation at 580.185 nm. An excited-state lifetime ($\tau_{obs}$) of 880 μs was obtained from a single exponential fit of the decay.

inhomogeneous line then reveals an increased transmission—that is, a spectral hole—at the burning frequency. For an excitation laser linewidth much narrower compared to the measured hole widths ($\Gamma_{hole}$), $\Gamma_h$ of a transition can be derived from the hole width ($\Gamma_h = \Gamma_{hole}/2$). The SHB method is important as it is used for tailoring absorption profiles in several quantum storage protocols and spin population initialization[21,44].

A spectral hole burned in the $^5D_0 \rightarrow {}^7F_0$ transition of [Eu$_2$] is shown in Fig. 3c. The hole was fitted by a Lorentz peak function, yielding a hole width of 43 ± 2 MHz at FWHM, corresponding to $\Gamma_h = 22 \pm 1$ MHz ($T_{2opt} = 14.5 \pm 0.7$ ns). The observed $\Gamma_h$ for [Eu$_2$] is relatively large compared to Eu(III) ions dispersed in matrices[40]. However, the $\Gamma_h$ is about an order of magnitude lower than homogeneous linewidths observed for Cr(III) complexes diluted in amorphous host lattices[39,45] and comparable to the ones reported for nitrogen-vacancy centers in diamond[46].

A lifetime-limited homogeneous linewidth of 180 Hz was calculated for [Eu$_2$], using the experimentally observed excited-state lifetime $\tau_{obs} = 880$ μs that includes local field correction effects[47,48]. The homogeneous linewidth of 22 ± 1 MHz obtained from the SHB studies is, therefore, much larger than the lifetime limited homogeneous linewidth. We attribute this additional broadening to fluctuations in Eu(III) environment caused by molecular vibrations and rotations, as in Cr(III) complexes[45]. Nuclear spin containing elements (H and Cl) surrounding the Eu (III) sites[49], as well as spectral diffusion arising due to disorder, defects, and grain boundaries[45,50–52] could also have contributed to dephasing (see Supplementary Section 3.3.5). More insights on the dephasing processes could be obtained by performing temperature-dependent studies[27,52], especially below 1 K, where

dephasing processes should be significantly reduced. However, experiments below 1 K require highly specialized equipment, namely dilution refrigerators using He$_3$/He$_4$ mixtures with optical access. Although studying dephasing under these conditions is useful, it is out of the scope of the present study.

To further confirm that the observed spectral holes result from population transfer and storage in a different spin level, several criteria were verified. First, the determined minimum time delay (5 ms) between hole burning and hole readout pulses was much longer than the excited-state lifetime of 880 μs, ensuring that no population remained in the optical excited state (Supplementary Figure 3). Second, we confirmed that holes could be erased on demand by scanning the laser over a frequency of 200 MHz with high intensity. These sweeping laser pulses excite all the spin population in the ground state, including the populations present in the storage levels (Fig. 3b), because typical Eu(III) ground-state hyperfine splittings are in the range 25–100 MHz[53]. Thermal equilibrium population distribution is then reestablished following spontaneous relaxation from the excited state. Finally, we observed that holes could be erased and then burned again at a different frequency within the optical inhomogeneous line (Supplementary Figure 4), unambiguously elucidating selective and reversible optical addressing of a sub-ensemble of Eu (III) ions.

The decay of the hole depth was observed by increasing the delay between burning and readout pulses (Fig. 3d). This measurement provides insight into the relaxation times ($T_{1spin}$) for the nuclear spin levels of Eu(III). Two components are observed in the experimental hole decay. A relaxation time of 1.6 ± 0.4 s is obtained for the fast component by a single

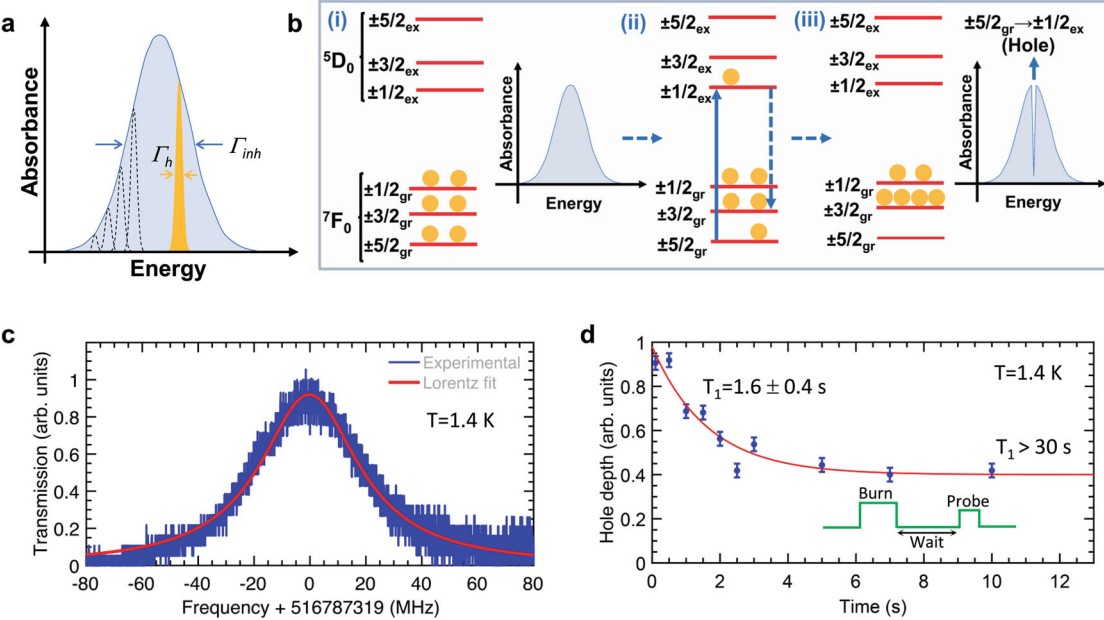

**Fig. 3 Spectral hole burning (SHB) in the $^5D_0 \rightarrow {}^7F_0$ transition of [Eu$_2$].** **a** An inhomogeneously ($\Gamma_{inh}$) broadened line is composed of narrow homogeneously ($\Gamma_h$) broadened lines. Selective optical excitation of one or several homogeneously broadened lines is used to burn spectral holes with implications for QIP applications. **b** Simplified mechanism of SHB. In the diagram, $\pm 1/2_{gr}$, $\pm 3/2_{gr}$, $\pm 5/2_{gr}$, and $\pm 1/2_{ex}$, $\pm 3/2_{ex}$, and $\pm 5/2_{ex}$ correspond to nuclear spin levels associated with the ground $^7F_0$ and excited $^5D_0$ states, respectively. (i) A laser scan reveals an inhomogeneously broadened absorption spectrum, like the one shown in (**a**), due to the excitation of multiple ions ensembles with infinitesimally small energy difference. (ii) Selective and continuous laser pumping transfers the population from one of the ground-state spin levels to an excited-state spin level. This is illustrated for ions with the $\pm 5/2_{gr}$ ($^7F_0$) to $\pm 1/2_{ex}$ ($^5D_0$) transition resonant with the laser. (iii) Population relaxation from level $\pm 1/2_{ex}$ to level $\pm 3/2_{gr}$, under the ambit of level $\pm 3/2_{gr}$ having sufficiently longer lifetime than levels $\pm 1/2_{gr}$ and $\pm 5/2_{gr}$, results in a decreased population in level $\pm 5/2_{gr}$. Since the spin level splittings are much lower than the optical inhomogeneous broadening, the laser will excite other ions along with transitions from the $\pm 3/2_{gr}$ or $\pm 1/2_{gr}$ levels, which will therefore be emptied. This finally results in a hole in the inhomogeneously broadened absorption spectrum. **c** Spectral hole burned in the $^5D_0 \rightarrow {}^7F_0$ transition of [Eu$_2$]. A Lorentzian fit reveals an FWHM of 43 ± 2 MHz, corresponding to $\Gamma_h = 22 \pm 1$ MHz. A $T_{2opt}$ = ~14.5 ns is calculated using the relation $\Gamma_h = (1/\pi T_{2opt})$. **d** The decay of the hole depth as a function of delay time before readout, showing the relaxation rate of the nuclear spin levels in the binuclear complex. **a** and **b** are partially reproduced with permission from ref. [23].

exponential fit. The second component is too slow to be accurately determined in the measurement time scale (10 s), but we estimate a lower bound relaxation time of 30 s. Multi-exponential hole decay curves have been previously observed in rare-earth materials[27], due to different spin–lattice relaxation rates among the ground-state spin levels. For the fast component, $T_{1spin}$ is not limited by spin flip-flops between neighboring Eu(III) ions since the intramolecular flip-flop rate is expected between $10^{-4}$ and $10^{-5}$ s$^{-1}$ [54], which is remarkably lower than the relaxation rate ($T_{1spin}^{-1}$) of ~0.6 s$^{-1}$ measured in the complex. Still, this relaxation time is more than three orders of magnitude larger than the excited-state lifetime; therefore, efficient spin population transfer can be achieved, enabling, for example, spin-state initialization.

## Discussion

The foregoing experimental observations demonstrate that REI-based molecular complexes could be utilized as photonic quantum materials for implementing QIP schemes. On a comparative scale, the magnitude of optical coherence lifetime ($T_{2opt}$) observed for [Eu$_2$] is several orders smaller than the values reported for Eu-doped systems. Unlike the doped systems, molecular systems offer the possibility to tune the optical properties via molecular engineering strategies. For example, ideal Eu(III) complexes showing long optical and spin coherence times could be obtained by designing ligands with deuterated backbones and nuclear spin-free donor sites and complexing

such ligands with isotopically enriched REIs. The molecular engineering strategy could also be adapted to obtain Eu(III) complexes with higher PL quantum yields than the 38 ± 6% observed for [Eu$_2$] (see the discussion in Supplementary Section 3.3.1). On the flip side, designing nuclear spin-free ligands or complete deuteration of structurally complex ligand backbones is a challenging task. Moreover, nuclear spins are difficult to address, rendering the qubit scalability based on molecular self-assembly a demanding task to achieve—synthesis of multinuclear REI systems is a strategy to overcome this issue[31].

In conclusion, long-lived spectral holes have been prepared in the inhomogeneously broadened $^5D_0 \rightarrow {}^7F_0$ optical transition of [Eu$_2$] at 1.4 K. The hole width corresponds to a homogeneous linewidth ($\Gamma_h$) of 22 ± 1 MHz, which is equivalent to an optical coherence lifetime ($T_{2opt}$) of 14.5 ± 0.7 ns. The burn mechanism is attributed to population redistribution within the ground-state nuclear spin levels of Eu(III) that act as shelving levels. The observation of narrow optical inhomogeneous linewidth and nuclear spin-state polarization associated with [Eu$_2$] relies on the presence of narrow $^5D_0 \rightarrow {}^7F_0$ optical transition in the complex. To progress towards quantum technology applications, REI-based molecular systems featuring optical coherence lifetime larger than the $T_{2opt}$ = 14.5 ± 0.7 ns reported for [Eu$_2$] in this study should be obtained. The tunable nature of quantum properties of molecules with precision could provide access to REI molecule-based quantum materials with technologically relevant spin and optical coherence lifetimes.

## Methods

Experimental descriptions on SHB spectroscopy, preparation of the complex, X-ray crystallography, and powder X-ray diffraction analysis of the complex are detailed in the supporting information associated with this article.

**Steady-state photoluminescence spectroscopy**. Steady-state emission spectra were recorded on an FLP920 spectrometer from Edinburgh Instrument working with a continuous 450 W Xe lamp and a red-sensitive Hamamatsu R928 photo-multiplier in Peltier housing. The emission spectrum of the europium complex in the near-IR (NIR) region was also measured by using a nitrogen-cooled Hama-matsu R5509-72 Vis-NIR detector (300–1700 nm), affording similar results. All spectra were corrected for the instrumental functions. When necessary, high pass filters at 330, 395, 455, or 850 nm were used to eliminate the second-order artifacts.

Fluorescence lifetimes were measured at room temperature on the same instrument working in the multi-channel spectroscopy mode and using a Xenon flash lamp as the excitation source. Errors on lifetimes are ±10%. Luminescence quantum yields were measured with a G8 Integrating Sphere (GMP SA, Switzerland) according to the absolute method detailed in ref. [55]. Estimated errors are ±15%.

**Low-temperature high-resolution spectroscopy and SHB measurements**. PLE, fluorescence lifetime, and SHB measurements were carried out on micro-crystalline powder (Supplementary Figure 1). Five milligrams of total material was added into a home-built brass container with front and rear optical access. The filled container was then introduced into a He bath cryostat (Janis SVT-200). The sample temperature was regulated by acting on the liquid He volume and pressure in the sample chamber, and it was continuously monitored by a Si diode (Lakeshore DT-610). Optical excitation was carried out by a tunable CW dye laser (Sirah Matisse DS) with 250 kHz linewidth. The excitation beam was focused onto the sample by a 75-mm-diameter lens and transmitted scattered light was collected by several lenses. Signals were detected with an avalanche photodiode (Thorlabs 110 A/M). The experimental setup is schematically represented in Supplementary Figure 2.

PLE spectra were recorded by scanning the excitation wavelength ($\lambda_{exc.}$) between 580.08 and 580.35 nm (vac.) while monitoring the fluorescence intensity from the $^5D_0$ excited state to the $^7F_J$ levels ($J = 2-6$). A long-pass filter (cut-off at 590 nm) was placed in front of the detector to reject the excitation wavelength. Low-temperature emission spectra were recorded under different excitation wavelengths using a thermoelectrically cooled CCD spectrometer (Avantes, AvaSpec-2048). The crystalline material showed no evidence of photobleaching during low-temperature experiments, which was confirmed by monitoring fluorescence intensity under CW laser excitation (Supplementary Figure 14).

SHB spectra and fluorescence decays were recorded by modulating the CW output of the laser in frequency and amplitude with an acousto-optic modulator (AOM; AA Optoelectronic MT200-B100A0, 5-VIS, 200 MHz central frequency), set in the double-pass configuration, and driven by an arbitrary waveform generator with 625 MS s$^{-1}$ sampling rate (Agilent N8242A). The $^5D_0$ fluorescence decay was recorded after a single excitation pulse of 2 ms at 580.185 nm. The SHB sequence was formed of ten burning pulses of 2 ms length, with an excitation power of 30 mW and at a fixed excitation wavelength within the inhomogeneous absorption profile. The waiting time between pulses was set to 5 ms to enable spontaneous relaxation from the excited state to the ground-state nuclear spin levels before a new burn. The spectral hole was then probed by monitoring the transmission of a frequency scanning pulse of 2 ms duration, with a frequency span of 200 MHz and excitation power of 5 mW. The recorded transmission under burn conditions was corrected from transmission obtained without burning, to cancel out the frequency-dependent response of the AOM. The delay before readout was varied from a minimum of 5 ms to a maximum of 10 s. Every SHB sequence was ended by a series of high-power pulses scanning over 200 MHz, to reset the ground-state population back to equilibrium. Spectral holes were recorded after averaging 50 sequences to improve the signal-to-noise ratio, for a better estimation of the hole width. The spectral width of the 2-ms-long burning pulse was limited by the laser linewidth (250 kHz), which is negligible compared to the measured spectral holes and also typical Eu(III) hyperfine splittings[53]. Hole linewidths were determined by Lorentz fit to the experimental data. Fluorescence lifetimes were determined by a single exponential fit to the decay data. The relaxation time for the fast component of the hole decay was also determined by a single exponential fit. The slow component lower bound limit was determined as the maximum decay possible in a time delay of 5 s within the experimental error bars given for the hole depths.

## Data availability

X-ray crystallographic data of the [Eu$_2$] and [Gd$_2$] complexes, in the form of cif files, discussed in this article can be obtained from the Cambridge Crystallographic Data Centre—CCDC 1863020 [Eu$_2$] and 1961893 [Gd$_2$]. The original data sets corresponding to the experiments can be obtained from the authors upon a reasonable request (contact: mario.ruben@kit.edu).

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

## Acknowledgements

M.R. thanks the grant agencies innovation FRC for the financial support for the project Molecular *Qudits*: Isotopologues Coordination Chemistry and the DFG priority program 1928 "COORNETS" for generous support. D.S. and P.G. have received funding from the European Union's Horizon 2020 research and innovation program under grant agreement no. 820391 (Square). A.M.N. and L.J.C. thank the French Agence National de la Recherche for financial support (Neutrino project no. ANR-16-CE09_0015-02). We all thank Dr. Asato Mizuno for the structural refinement of [Gd₂] complex.

## Author contributions

M.R. and P.G. conceived and supervised the project. K.S.K. synthesized and characterized the complex and involved in the conceptual development of the project. L.K. determined the X-ray structure of the complex. B.H. performed powder X-ray diffraction studies and indexed the patterns. A.M.N. and L.J.C. carried out the steady-state photoluminescence studies. D.S. and P.G. performed PLE and SHB measurements. K.S.K., D.S., P.G., and M.R. wrote the manuscript. All the authors have read and commented on the manuscript.

## Funding

## Competing interests

The authors declare no competing interests.
