## [Peer Review File · Nature Communications]

Reviewers' Comments:

Reviewer #1:

Remarks to the Author:

Most of the previous comments and suggested corrections have been addressed. Unfortunately, the suggested additional experiments have not been performed.

Regarding "More insights on the dephasing processes could be obtained by performing temperature-dependent studies, especially below 1 K, where dephasing processes should be significantly reduced." It is a pity that the authors rejected to include such important studies in the paper which definitely would have made the manuscript stronger. At the minimum, it would be good to include after the above sentence that studies below 1 K are not within the scope of this contribution as those are rather time-consuming and elaborate. This is better than just postulating as this confuses the reader and raises the question why such studies have not been done then if those would have been so useful.

Regarding "The usefulness of such a transition for QIP applications could be enhanced by means of molecular engineering approaches. The tunable nature of molecular properties with atomic precision could provide access to quantum materials with technologically relevant spin and optical coherence lifetimes." These two sentences read too vague and cannot be backed-up currently as there are no molecular Eu systems known yet that have been implemented in applications. In addition, the optical coherence time (T_{2opt}) determined for [Eu2] in this current manuscript is multiple orders of magnitude smaller than what has been observed in europium-doped system. Thus, the magnitude of the optical coherence time observed for [Eu2] is not special here, only the fact that an optical coherence time has been observed for the first time in a molecular system. The next logical step would be to design molecules that show optical coherence times comparable to or ideally even better than what has been demonstrated in europium-doped systems. Thus, I suggest rephrasing both sentences and lessening the "usefulness" reasoning as I think there is a long way to go until something like that could be really implemented for QIP applications.

After addressing my relatively minor comments, the manuscript may be considered for publication in *Nature Communications*.

Answers to the comments

Note: The new additions are highlighted in yellow in the revised script. The revised script was edited by switching on the track change mode ON. The corrections could be seen by putting the script in the “All Markup” view in the review panel of MS Word.

Comment: Most of the previous comments and suggested corrections have been addressed.

Answer: We thank the referee for the constructive criticism and comments, which helped us to improve the quality and presentation of the script.

Comment: Unfortunately, the suggested additional experiments have not been performed. Regarding “More insights on the dephasing processes could be obtained by performing temperature-dependent studies, especially below 1 K, where dephasing processes should be significantly reduced.” It is a pity that the authors rejected to include such important studies in the paper which definitely would have made the manuscript stronger. At the minimum, it would be good to include after the above sentence that studies below 1 K are not within the scope of this contribution as those are rather time-consuming and elaborate. This is better than just postulating as this confuses the reader and raises the question why such studies have not been done then if those would have been so useful.

Answer: We have added the following sentence, as suggested by the referee:

“However, experiments below 1 K require highly specialized equipment namely dilution refrigerators using He₃/He₄ mixtures with optical access. Although studying dephasing under these conditions is useful, it is out of the scope of the present study.”

Comment: Regarding “The usefulness of such a transition for QIP applications could be enhanced by means of molecular engineering approaches. The tunable nature of molecular properties with atomic

precision could provide access to quantum materials with technologically relevant spin and optical coherence lifetimes.” These two sentences read too vague and cannot be backed-up currently as there are no molecular Eu systems known yet that have been implemented in applications. In addition, the optical coherence time (T_{2opt}) determined for [Eu₂] in this current manuscript is multiple orders of magnitude smaller than what has been observed in europium-doped system. Thus, the magnitude of the optical coherence time observed for [Eu₂] is not special here, only the fact that an optical coherence time has been observed for the first time in a molecular system. The next logical step would be to design molecules that show optical coherence times comparable to or ideally even better than what has been demonstrated in europium-doped systems. Thus, I suggest rephrasing both sentences and lessening the “usefulness” reasoning as I think there is a long way to go until something like that could be really implemented for QIP applications.

After addressing my relatively minor comments, the manuscript may be considered for publication in Nature Communications.

Answer: We have rephrased the sentence following the referee comment:

“To progress towards quantum technology applications, REI-based molecular systems featuring optical coherence lifetime larger than the $T_{2opt} = 14.5 \pm 0.7$ ns reported for [Eu₂] in this study should be obtained. The tunable nature of quantum properties of molecules with precision could provide access to REI-molecule-based quantum materials with technologically relevant spin and optical coherence lifetimes.”